# GCNFT: Graph Convolutional Networks Aware Generative Feature Transformation

## Abstract

Feature transformation for attributed graphs converts raw node attributes into augmented features that preserve node and structure information. Relevant literature either fails to capture graph structures (e.g., manual handcrafting, discrete search), or is latent and hard to interpret (e.g., GCNs). How can we automatically reconstruct explicit features of an attributed graph while effectively integrating graph structures and attributes? We generalize the learning task under such setting as a GCN-aware Feature Transformation (GCNFT) problem. GCNFT imposes two under-addressed challenges: 1) quantifying GCN awareness and 2) bridging GCN awareness and feature transformation. To tackle these challenges, we propose a graph convolution structure score guided generative learning framework to solve GCNFT. To quantify GCN awareness, we interpret GCN as a gap minimization process between ideal and current node representations in iterative Laplacian smoothing, and develop a task-agnostic structure score to approximate GCN awareness. To incorporate GCN awareness, we model feature transformation as sequential generative learning so that we pave a way to leverage the structures score to guide the generative learning and encourage graph structure alignment. Extensive experiments demonstrate the proposed GCN-aware approach outperforms feature transformation baselines with an improvement of 3% to 20% over node, link, and graph prediction tasks. Our code is available at https://anonymous.4open.science/r/GCNFT.

## 1 Introduction

In many attributed graph systems (e.g., social network analysis, financial network fraud detection, graph recommendations), raw attributes alone may not reveal hidden structural interconectivity patterns. For instance, simply using raw attributes (e.g., user age, product category) often fails to capture user-product interactions. Feature transformation for attributed graph is to convert raw attributes of nodes in a graph into new, more informative features for graph machine learning. It can represent complex relationships, augment information of nodes with sparse connections, create a more homogeneous feature space from heterogeneous types of nodes and attributes, and capture more general properties of graph structures.

Relevant literature is two fold: 1) Graph Neural Networks (GNNs) and latent transformed representations: GNNs have emerged as powerful tools for encoding graph topology and node attributes into a lower-dimensional latent space. Key architectures include Graph Convolutional Networks (GCNs), Graph Attention Networks (GATs), and GraphSAGE. However, their output latent representations are often hard to interpret, making it challenging to understand what specific aspects of graph structure or node attributes contribute to the feature space. 2) classic feature transformation and explicit transformed representations: classic feature transformation involves creating handcrafted features such as node centrality, clustering coefficients, community membership, and attribute aggregation to represent graph structure and node attributes for downstream tasks. Despite their interpretability, these methods require domain knowledge and manual effort, cannot model complex non-linear relationships, are prone to overfit, and cannot scale up. Recently,

automated feature transformation methods (Uddin et al., 2021; Ying et al., 2024) are designed for generic tabular data, and they ignore the topological structure in attributed graphs.

This gap inspires the problem of *GCN-aware Feature Transformation (GCNFT)* that answers: how can we automatically transform and reconstruct explicit feature space of an attributed graph, described by a node feature matrix and an adjacency matrix, while effectively integrating graph structures and attributes?

There are two major challenges in solving GCNFT: 1) quantifying GCN awareness; 2) bridging GCN awareness and feature transformation. Firstly, GCN is a multi-step process: graph convolution operation, node attribute transformation, stacking layers for multi-hop aggregation, and finally optimized by a task-specific objective. We need to approximate GCN awareness into a task-agnostic optimizable regularization term to encourage feature transformation to pay attentions to graph structures. Secondly, classic feature transformation are usually based on empirical handcrafting or discrete search methods. Direct incorporation of GCN awareness regularization term into such methods will fail due to the non-differentiable, discrete nature of the search space, and the potential disruption of heuristic evaluations and algorithmic stability. We need to model feature transformation as a modern learnable framework (data, model, objective, optimization) so as to bridge the gap between GCN awareness and feature transformation.

**Our insights: leveraging graph convolutional structure score guided GenAI to unify GCN awareness and feature transformation into an optimizable learning paradigm.** Classic feature transformation methods are usually non-differentiable and discrete search based. The success of LLMs showcases the abilities of GenAI to encode discrete sequential tokens into continuous representations, and generate actionable responses. Follow the similar spirit, we view a sequence of feature transformation operations (e.g., $f_1 * f_2, f_2/f_3, \sqrt{f_4}, ...$) as a generated token sequence. We highlight that attributed graph feature transformation can be formulated as a sequential token generation task within a continuous space. In this context, the transformed feature set is expressed as an embedding vector, which is subsequently optimized and generated by an encoder-decoder framework. We prove that GCN awareness can be theoretically approximated by a graph convolutional structure score to guide the generative process, where feature transformations are not just downstream task-optimized but also graph structure-preserving.

**Summary of technical solution.** Inspired by these insights, we develop a graph convolution aware generative feature transformation method for attributed graphs. Our method includes two components: 1) quantifying GCN awareness: we approximate the graph convolution operation, non-linear transformation, and multi-hop aggregation of GCN into a graph convolutional structure score as a regularization term. 2) GCN-aware transformation generation: we develop a graph convolutional structure score guided generative feature transformation approach that consists of embedding (i.e., to learn a GCN-aware feature transformation embedding space), optimization (i.e., to identify the embedding point for the best GCN-aware transformation), and generation (i.e., to decode the optimal GCN-aware transformation operation sequence).

**Our contributions** are: 1) *AI task:* We formulate a novel task of graph convolution aware explicit feature transformation for attributed graphs. 2) *Framework:* We develop a graph convolution structure score guided generative learning framework to generate task-optimized and structure-preserving attributed graph feature transformations. 3) *Computing:* An approximation technique with mathematical inference is developed to reduce GCN into a task-agnostic structure loss term. We convert attribute graph feature transformation into differentiable generative learning to enable the incorporation of GCN awareness to steer optimization.

## 2 Definitions and Problem Statement

### 2.1 Important Definitions

**Operations & Crossed Feature.** We apply two types of mathematical transformations on attributes: 1) unary operations:'$log, sin$', and etc; 2) binary operations: '$+, -$', and etc. By crossing features within an

attribute graph, we reconstruct the representation space to enhance the data's AI capability. For instance, selected transformations generate new features (e.g., $\mathbf{f}_1 + \mathbf{f}_2, sin(\mathbf{f}_2) - exp(\mathbf{f}_3)$).

**Feature Cross Sequence.** Treating a feature/operation as a symbol allows the pattern of these symbols to represent the knowledge inherent in crossed features. We use symbolic expressions to represent crossed features, which enables us to assess their quality without directly involving the original data (e.g., '[sos][$\mathbf{f}_1$][+][$\mathbf{f}_2$][sep][$sin$][(][$\mathbf{f}_2$][)][−][$exp$][(][$\mathbf{f}_3$][)][eos]').

## 2.2 The GCNFT Problem

Given an attribute graph $\mathcal{G} = (A, X, y)$, where $A$ is an adjacency matrix, $X = [\mathbf{f}_1, \mathbf{f}_2, ..., \mathbf{f}_n]$ is an attribute matrix, in which a row represents a node and a column represents a kind of attribute (a.k.a, feature), and $y$ is the target labels of nodes, edges or graphs. Our goal is to automatically reconstruct an optimal and explicit representation space $X^*$, which captures structure relationships achieved by Graph Convolutional Networks $G$ and improves a downstream ML task $\mathcal{M}$ (e.g., node classification, link prediction, or graph classification):

$$X^* = \arg \max_{\hat{X}}(\mathcal{I}_\mathcal{M}(\hat{X}, y) + \mathcal{S}(\hat{X}, G(\mathcal{G})), \tag{1}$$

where $\hat{X}$ is a generated attribute matrix that includes multiple original features and crossed features, $X^*$ is the best transformed attribute matrix, $\mathcal{I}$ is the predictive performance indicator, $\mathcal{S}(\hat{X}, G(\mathcal{G}))$ is the structure score that measures how well $\hat{X}$ fits the structure relationships of $G(\mathcal{G})$.

# 3 Graph Convolution Aware Generative Feature Transformation

## 3.1 Overview of Proposed Method

Figure 1 shows our solution includes two phases: 1) quantifying GCN awareness; 2) GCN-aware attributed graph feature transformation generation. Specifically, in Phase 1, we leverage the iterative Laplacian smoothing concept to infer ideal node representation and develop an approximation of GCN awareness by the similarity score between ideal and current note representations. This structure score is a generic, task-agnostic and optimizable regularization term for diverse machine learning paradigms. In Phase 2, we develop a graph convolution structure score guided encoder-decoder approach for generative attributed graph transformation. This approach can impose GCN awareness to guide the encoder (how to map a feature transformation into an embedding vector) and the decoder (how to search the best feature transformation embedding point for generation).

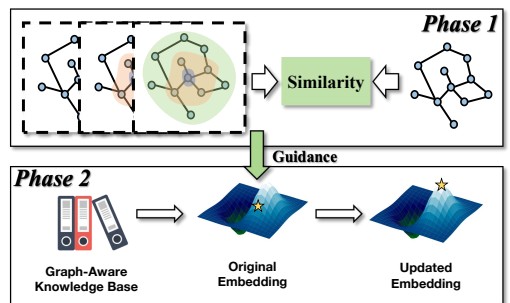

Figure 1: Framework Overview

## 3.2 Quantifying GCN Awareness as Task-agnostic Convolutional Structure Score

**Why Quantifying GCN Awareness Matters?** In attributed graphs, node features interact closely with graph structure to encode graph connectivity patterns in an attribute space. Classic feature transformations on tabular data ignore underlying graph structures, and, thus, fail to leverage these relationships, leading to suboptimal representations. Since GCNs excel at capturing structural relationships through aggregation and propagation, it is appealing to incorporate GCN awareness into classic feature transformations, in order to align feature transformation with graph structures. Therefore, GCN awareness quantification can provide a measurable and optimizable mechanism to ensure that transformed features are structurally consistent.

**Leveraging GCN as the Gap Minimization between Ideal and Current Node Representations in Iterative Laplacian Smoothing to Make GCN Awareness Computationally Tangible.** We find that the

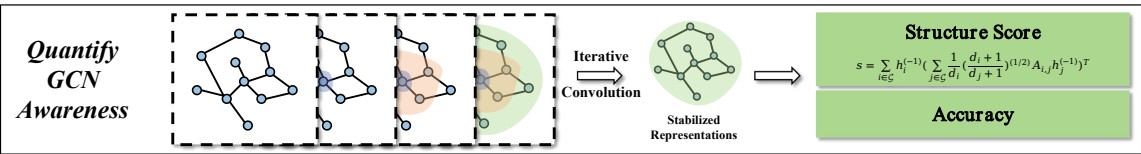

Figure 2: Quantify GCN Awareness

aggregation and propagation mechanism of GCN can be seen as a recurrent process of minimizing the gap between ideal node representation and current node representation in iterative Laplacian smoothing. Inspired by this finding, we propose to quantify the degree of similarity between the ideal and current node representations as an indicator of GCN effects in feature transformation by a two step method:

**Step 1: Identifying Ideal Node Representation.** The core operation of GCN involves aggregating node features via the adjacency matrix, while Laplacian smoothing diffuses node features to their neighbors using the graph's Laplacian matrix, making node features more similar. The work by (Kipf & Welling, 2016) demonstrates that the propagation mechanism of GCN can be understood as an iterative application of Laplacian smoothing, which is formally expressed by the following equation: $\mathbf{H}^{(l+1)} = \sigma(\widetilde{\mathbf{D}}^{-\frac{1}{2}}\widetilde{\mathbf{A}}\widetilde{\mathbf{D}}^{-\frac{1}{2}}\mathbf{H}^{(l)}\mathbf{W}^{(l)})$, where $\widetilde{\mathbf{D}}^{-\frac{1}{2}}\widetilde{\mathbf{A}}\widetilde{\mathbf{D}}^{-\frac{1}{2}}$ is the Normalized Laplacian matrix, in which $\widetilde{\mathbf{A}}$ is the adjacency matrix of the graph with self-loops added and $\widetilde{\mathbf{D}}^{-\frac{1}{2}}$ is the degree matrix corresponding to $\widetilde{\mathbf{A}}$, $\mathbf{H}^l$ represents the node feature matrix at layer $l$, $\mathbf{W}^{(l)}$ is the weight matrix for layer $l$, and $\sigma$ is the activation function.

As GCN repeatedly applies this smoothing operation, the attribute in the final layer gradually becomes smoother through iterative updates. Eventually, the node representations stabilize in the graph structure, and once the output no longer changes significantly with further iterations, the model is considered to be converged. The ideal node representation in the final layer of GCN is: $\mathbf{H}^{(-1)} = \sigma(\widetilde{\mathbf{D}}^{-\frac{1}{2}}\widetilde{\mathbf{A}}\widetilde{\mathbf{D}}^{-\frac{1}{2}}\mathbf{H}^{(-1)}\mathbf{W}^{(-1)})$, where $\mathbf{H}^{(-1)}$ is the node representation matrix in the final layer of GCN.

In the simplification discussed in (Veličković et al., 2018; Wu et al., 2019), to further reduce the complexity of GCN's final layer representation, the non-linear activation function and the weight matrix can be removed, so the above equation is approximated as: $\mathbf{H}^{(-1)} = \widetilde{\mathbf{D}}^{-\frac{1}{2}}\widetilde{\mathbf{A}}\widetilde{\mathbf{D}}^{-\frac{1}{2}}\mathbf{H}^{(-1)}$. More specifically, the ideal solution for the final representation of each node in the graph can be represented as:

$$\mathbf{h}_i^{(-1)} = \sum_{j\in\mathcal{G}} \frac{1}{\sqrt{(d_i+1)(d_j+1)}}\mathbf{A}_{i,j}\mathbf{h}_j^{(-1)} + \frac{1}{d_i+1}\mathbf{h}_i^{(-1)} = \sum_{j\in\mathcal{G}} \frac{1}{d_i}\sqrt{\frac{d_i+1}{d_j+1}}\mathbf{A}_{i,j}\mathbf{h}_j^{(-1)}. \tag{2}$$

In this context, $\mathcal{G}$ refers to the given graph, $\mathbf{h}_i$ denotes the representation of any node within the graph, $\mathbf{A}_{i,j}$ corresponds to the adjacency matrix of the graph, and $d_i$ is the degree of node $i$.

**Step 2: Measuring the Similarity between Ideal and Current Node Representation as Structure Score.** Based on the above analysis, we can train a relaxed GCN model by guiding node representations to approximate ideal node representations estimated in Equation 2, as well as optimizing the performance in downstream tasks. We defined a structure score to estimate how closely current node representations match ideal node representations, which is denoted as the similarity between them. Here we exploit the cosine similarity as the similarity metric:

$$s = \sum_{i\in\mathcal{G}} \frac{\mathbf{h}_i^{(-1)}(\sum_{j\in\mathcal{G}} \frac{1}{d_i}\sqrt{\frac{d_i+1}{d_j+1}}\mathbf{A}_{i,j}\mathbf{h}_j^{(-1)})^\top}{\|\mathbf{h}_i^{(-1)}\|\|\sum_{j\in\mathcal{G}} \frac{1}{d_i}\sqrt{\frac{d_i+1}{d_j+1}}\mathbf{A}_{i,j}\mathbf{h}_j^{(-1)}\|} \tag{3}$$

To simplify the equation, we conduct vector normalization on the learned representations, and thus each representation $\mathbf{h}_i^{(-1)}$ has a similar l2-norm. As a result, $s$ is equivalent to:

$$s = \sum_{i\in\mathcal{G}}\mathbf{h}_i^{(-1)}(\sum_{j\in\mathcal{G}} \frac{1}{d_i}\sqrt{\frac{d_i+1}{d_j+1}}\mathbf{A}_{i,j}\mathbf{h}_j^{(-1)})^\top \tag{4}$$

Inspired by this analysis, we can integrate GCN awareness into feature transformation by evaluating the structure score of a transformed attribute matrix. The evaluation feedback can enforce feature transformation to better align with the structural characteristics of GCN.

### 3.3 GCN-AWARE GENERATIVE FEATURE TRANSFORMATION FOR ATTRIBUTED GRAPHS

**Why Using A Knowledge Guided Generative Learning Perspective to Bridge Feature Transformation and GCN Awareness.** After deriving graph convolutional structure score as the regularization term of GCN awareness, we need to incorporate the structure score into feature transformation to align graph attributes with graph topology. The key challenge of incorporating GCN awareness is that classic feature transformation methods are based on empirical handcrafting and discrete search, thus, there is no way to incorporate the GCN awareness into attributed graph feature transformation. To address this issue, we need to frame feature transformation as a modern optimizable learning paradigm (data, model, objective, optimization), so that the GCN awareness can serve as knowledge guidance to guide feature transformation learning. We regard a transformed feature set (e.g., $f_1 * f_2, f_2/f_3, \sqrt{f_4}, ...$) as a token sequence, the search of the best transformed feature set as the generation of the maximized likelihood token sequence, so attributed graph feature transformation is reformulated as a generative learning paradigm of encoding, optimization, and decoding. This reformulation provides an opportunity for the graph convolutional structures score to guide the generative learning and enforce the alignment with graph structure.

**Leveraging Graph Convolutional Structure Score to Guide Generative Feature Transformation Learning.** This GCN-aware generative learning paradigm includes three steps: 1) we firstly embed GCN-aware feature transformations into embedding vectors and reconstruct feature cross sequences, by optimizing not just sequential reconstruction and a predictive accuracy estimator, but also a structure score estimator; 2) once the model converges, the GCN-aware estimators guide the search of the optimal embedding point with the highest structure score and downstream task performance. 3) we finally decode the optimal embedding point to generate the best feature cross sequence that is converted into the optimal transformed feature space by predefined rules. We next detail its data, model, optimization, and generation components.

**1) Data: GCN-aware Attributed Graph Feature Transformation Knowledge Acquisition.** Inspired by (Wang et al., 2022a), we develop a reinforcement learning system to automatically explore and collect various transformations of graph node attribute matrices as a knowledge base for training the generative model, as shown in Figure 3. The reinforcement exploration experiences (a.k.a, transformed attribute graph feature sets), along with corresponding structure scores and downstream task accuracy, are formatted as training data. The reinforcement learning system includes: 1) Multi-Agents: We design three agents—a head feature agent, an operation agent, and a tail feature

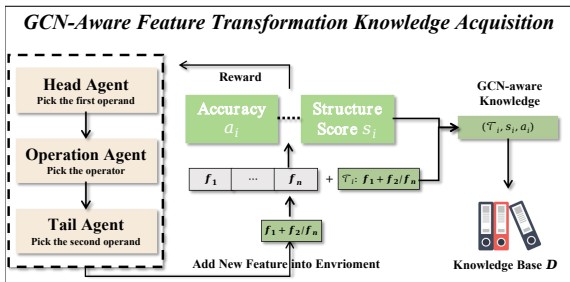

Figure 3: Integrate GCN Awareness into Generative Feature transformation

agent—to perform feature crossing. 2) Actions: At each reinforcement iteration, the agents select a head feature, an operation, and a tail feature to generate a new feature, which is added to the feature set for future iterations. 3) Reward Function: This exploratory data collection process is incentivized by a reward of not just downstream performances but also structure scores to collect high-quality training data with similar GCN-like transformation patterns. As the agent policies grow, we can collect many high-quality transformed attributed graphs with corresponding structure scores $s$ and accuracy $a$. We regard feature and operation as tokens, and then use postfix expression to convert the transformed attributed matrix as feature token sequences $\mathcal{T}$ (e.g., $f_1 + f_2, sin(f_2) \rightarrow$'[sos][$f_1$][$f_2$][+][sep][$f_2$][$sin$][eos]'). These three types of data construct the GCN-aware knowledge base $D = (\mathcal{T}_i, s_i, a_i)_{i=1}^{k}$, where $k$ indicates the number of samples.

**2) Model: Sequential Autoencoder.** We use the GCN-aware knowledge base $D = (\mathcal{T}_i, s_i, a_i)_{i=1}^k$ for training an encoder $\phi(\cdot)$ and a decoder $\psi(\cdot)$ to construct a latent embedding space as shown in Figure 4. We utilize a single layer Long Short-Term Memory (LSTM) (Hochreiter & Schmidhuber, 1997) network for the encoder to obtain latent embedding of the feature token sequence, denoted by $\mathbf{e} = \phi(\mathcal{T})$. We also use a single layer LSTM for a decoder to reconstruct the feature token sequence. Given a latent embedding $\mathbf{e}$, we leverage the negative log-likelihood of the distribution of the decoder's output $P_\psi$ to measure the difference between the generated token sequence and the real one, defined as $\mathcal{L}_{rec} = -logP_\psi(\mathcal{T}|\mathbf{e})$.

**3) Optimization: Incorporating GCN Awareness into Embedding Space Learning and Optimal Embedding Point Search.** To generate the optimal feature token sequence, we first evaluate the latent embedding space for targeted optimization. We develop two evaluators to assess the relationship between the latent embeddings, the GCN structural awareness, and downstream task performance. The structural evaluator $\kappa(\cdot)$ estimates the correlation between latent embeddings and GCN awareness, which provides constraints to ensure the structural properties of the generated feature token sequences. The performance evaluator $\vartheta(\cdot)$ measures the relationship between the latent embeddings and downstream performance, which offers optimization targets to improve latent embeddings for better downstream outcomes. We iteratively optimize these two constraints to search for latent embeddings that possess both GCN structural awareness and enhanced downstream performance.

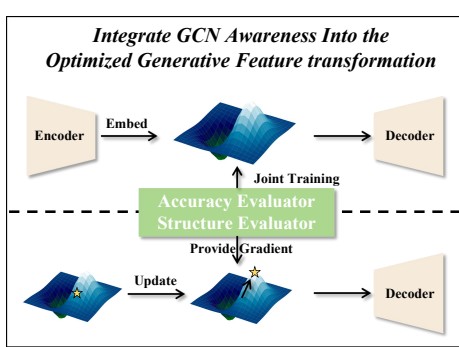

Figure 4: GCN-Aware Feature Transformation Knowledge Acquisition

*(i)* GCN Awareness Evaluator. We expect the latent embeddings to represent the structural score of the corresponding feature token sequence, which enables us to purposefully optimize the latent embeddings to obtain a feature token sequence with a higher structural score (i.e., where the generated node representations are closer to the ideal node representations aggregated by GCN in structure). We construct the relationship using a GCN awareness evaluator regressor function $\kappa(\cdot)$, and the estimated structure score is denoted by: $\hat{s} = \kappa(\mathbf{e})$. We use a simple linear layer to implement this evaluator function. We train the parameters of the evaluator by minimizing the Mean Squared Error (MSE) between the estimated structure score and the ground truth $\arg\min \mathcal{L}_{str} = MSE(s, \hat{s})$.

*(ii)* Performance Evaluator. We expect the latent embedding space to indicate the accuracy of the corresponding feature token sequence in downstream tasks. So we can optimize the latent embeddings for higher performance. We establish this relationship using a performance evaluator function $\vartheta(\cdot)$, and the estimated accuracy is denoted by: $\hat{a} = \vartheta(\mathbf{e})$. We use a linear layer to implement this evaluator and train its parameters by minimizing the MSE between the estimated accuracy and true value $\arg\min \mathcal{L}_{per} = MSE(a, \hat{a})$.

*(iii)* Iterative Optimization of GCN Awareness and Performance for the Optimal Latent Embedding. After constructing the latent embedding space, we enable the gradient-steered optimal search to find better embeddings. To simulate GCN training (i.e., node aggregation and performance-guided optimization), we introduce a two-stage optimization method. Specifically, we first conduct structure score optimization to simulate node aggregation that tends to cluster connected nodes. To implement this process, we leverage the gradient of structure score to guide optimization. Formally, given a starting point $\mathbf{e}$, the embedding after structure score optimization is given by: $\mathbf{e}^+ = \mathbf{e} + \alpha\frac{\sigma[\kappa(\mathbf{e})]}{\sigma\mathbf{e}}$, where $\alpha$ is the step length. By varying $\alpha$, we can generate samples in different levels of clustering. We regard these generated samples as the outputs of GCN in different epochs. Thereafter, for the outputs, we conduct performance-guided optimization. The target embedding of an optimization step is expressed by: $\mathbf{e}^* = \mathbf{e}^+ + \beta\frac{\sigma[\vartheta(\mathbf{e}^+)]}{\sigma\mathbf{e}^+}$, where $\beta$ denotes the step length. For each $\mathbf{e}^+$, we perform multi-step search, to find the optimal embedding.

**4) Generation: Generating Feature Cross Sequences to Reconstruct Attributed Graph Features.** When the optimal embedding $\mathbf{e}^*$ is searched, we generate the feature cross sequence by the well-trained decoder. Assuming the token sequence $f_1...f_{d-1}$, we infer the next token $f_d$ by maximize likelihood estimation: $f_d = argmax(P_\psi(f_d|\mathbf{e}^*, [f_1, f_2, ..., f_{d-1}]))$, Feature tokens are autoregressively generated until the end token (EOS) is generated. Finally, the optimal attribute graph is generated through the predefined rules.

## 4 EXPERIMENTAL RESULTS

### 4.1 EXPERIMENTAL SETUP

**Data Description & Evaluation Metrics** We select 3 graph datasets from TUDataset[1]. Our method is evaluated from three perspectives: 1) node classification; 2) link prediction; 3) graph classification to demonstrate the advantages of our model in graph-related tasks. Table 1 shows the statistics of the datasets. We adopt a two-layer MLP as the downstream model to evaluate the performance of generated attribute graphs. We use the F-1 score to measure the accuracy of graph downstream tasks.

Table 1: Key statistics of the datasets.

| Dataset | Graphs | Classes | Avg. Nodes | Avg. Edges | Node Labels | Node Att. |
|---|---|---|---|---|---|---|
| ENZYMES | 600 | 6 | 32.63 | 62.14 | ✔ | ✔ |
| PROTEINS_full | 1113 | 2 | 39.06 | 72.82 | ✔ | ✔ |
| Synthie | 400 | 4 | 95.00 | 172.93 | ✗ | ✔ |

**Baseline Algorithms & Model Variants** We compare the proposed method with 8 widely-used algorithms: 1) **PCA** (Abdi & Williams, 2010) reconstructs the feature space according to the original feature set. 2) **ERG** expands feature space by applying operation on each feature and selects valuable features. 3) **LDA** (Blei et al., 2003b) obtains features based on matrix factorization. 4) **NFS** (Chen et al., 2019); 5) **RDG** randomly generates features to reconstruct the feature space. 6) **TTG** (Khurana et al., 2018) regards feature transformation as a graph and performs reinforcement learning-based search; 7) **GRFG** (Wang et al., 2022b) proposes a feature grouping strategy and employs three agents to generate new features; 8) **MOAT** (Wang et al., 2024) embeds feature transformations and generate new feature transformations by gradient-based search. Besides, to comprehensively evaluate the proposed framework, we introduce two variants: 1) *w/o* **SO** and 2) *w/o* **PO** denote the model without structure optimization and without performance optimization.

**Hyperparameter Setting & Experimental Environment** In the data collection stage, we explore 600 epochs of feature transformations. In the training stage, we set the weights of $\mathcal{L}_{per}, \mathcal{L}_{str}$, and $\mathcal{L}_{rec}$ as 0.5, 0.4, and 0.1 respectively. In the generation stage, we perform 2 steps of structure optimization and 4 steps of performance optimization. All experiments are conducted on the Ubuntu 22.04.3 LTS operating system, utilizing an Intel(R) Core(TM) i9-13900KF CPU@ 3GHz, along with a single RTX 6000 Ada GPU and 49GB of RAM. Experiments were performed under the framework of Python 3.10.14 and PyTorch 2.0.1.

### 4.2 RESULT ANALYSIS

**Overall Comparisons.** This experiment aims to answer: *Can the proposed method effectively improve downstream tasks compared with baselines?* Table 2 shows the overall comparison results on the 3 datasets. We observe that GCNFT achieves the best performance across all datasets and tasks, with an average improvement of 3% over the best baseline. This is because the existing methods perform feature transformation only on tabular data. Without graph information, they show limited performance on graph-related tasks. In contrast, our model can leverage graph information to enhance feature transformation, facilitating GCN awareness. Notably, in graph classification, our model significantly outperforms the baselines. The potential

---

[1] https://chrsmrrs.github.io/datasets/docs/datasets/

Table 2: Three types of graph tasks: node classification, link prediction, and graph classification on three datasets.

| Method | ENZYMES | | | PROTEINS_full | | | Synthie | | |
|---|---|---|---|---|---|---|---|---|---|
| | Node | Link | Graph | Node | Link | Graph | Node | Link | Graph |
| PCA | 75.63 | 58.18 | 16.57 | 83.43 | 94.18 | 43.80 | - | 50.49 | 18.00 |
| ERG | 80.97 | 57.14 | 15.47 | 83.87 | 92.91 | 55.53 | - | 49.52 | 21.99 |
| LDA | 68.53 | 56.34 | 8.85 | 68.42 | 69.38 | 53.71 | - | 49.85 | 10.00 |
| NFS | 77.58 | 58.13 | 9.33 | 84.74 | 94.84 | 59.81 | - | 50.58 | 23.51 |
| RDG | 78.81 | 58.02 | 15.47 | 83.41 | 95.01 | 61.39 | - | 50.27 | 22.66 |
| TTG | 82.03 | 56.58 | 16.08 | 88.16 | 94.56 | 61.60 | - | 50.80 | 27.53 |
| GRFG | 84.86 | 59.34 | 15.81 | _89.36_ | 96.30 | 62.66 | - | 50.58 | 37.62 |
| MOAT | _90.70_ | _62.95_ | _17.17_ | 86.23 | _97.90_ | _65.39_ | - | _51.63_ | _43.82_ |
| GCNFT | **94.01** | **65.96** | **21.87** | **91.08** | **98.75** | **70.42** | - | **52.02** | **49.49** |

driver is that the structure optimization process clusters the connected nodes, making the graph embedding more discriminative.

**A Study of Optimization Methods.** This experiment aims to answer: *Is the two-stage optimization method helpful to improve the performance?* To investigate the effectiveness of the two-stage optimization method, we introduce two variants of GCNFT: *w/o* SO and *w/o* PO, denoting models without structure score optimization and performance optimization respectively. As shown in Figure 5, solely conducting structure optimization or performance optimization falls into suboptimal results. This phenomenon can be explained by two factors: 1) Without structural optimization, the model loses the ability to leverage graph information. It only focuses on optimization for tabular data, making it difficult to achieve optimal results. 2) Without performance optimization, the model loses the supervised signals from downstream task feedback, preventing it from conducting a gradient search toward performance improvement. Combining the two processes, we enable GCN-aware feature transformation to aggregate nodes and optimize the performance.

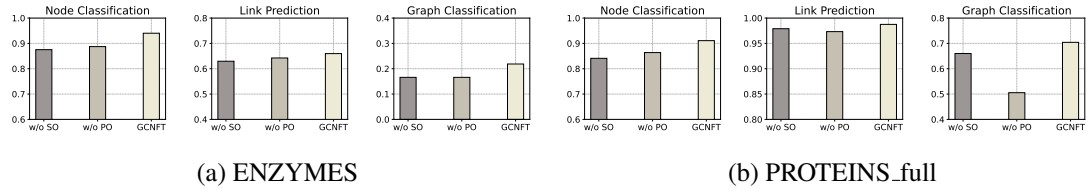

(a) ENZYMES                     (b) PROTEINS_full

Figure 5: Results of the proposed method with different optimization process, where *w/o* SO denotes the model without structure optimization, and *w/o* PO denotes the model without performance optimization.

**Robustness Check.** This experiment aims to answer: *Can the proposed model keep robust when different downstream models are used?* We investigate the performance of GCNFT on the ENZYMES dataset for the node classification task, utilizing Multi-Layer Perceptron (MLP) (Murtagh, 1991), K-Nearest Neighborhood (KNN) (Guo et al., 2003), Support Vector Machine (SVM) (Hearst et al., 1998), LASSO (Zou, 2006), and Ridge (Hoerl & Kennard, 1970) as downstream models respectively. We report the results in Table 3. The baselines are not robust and stable when collaborating with different downstream ML models. None of them can consistently achieve the second-best performance on this task. In contrast, GCNFT outperforms all the baselines with the highest improvement of 3.31%, regardless of the downstream ML model. This implies that GCNFT can formulate specific optimization strategies for different downstream tasks. Through a well-constructed feature embedding space, GCNFT performs a robust search towards the feature set with the best performance on specific downstream tasks.

Table 3: Different downstream ML models on the node classification task of ENZYMES.

| ML Model | PCA | ERG | LDA | NFS | RDG | TTG | GRFG | MOAT | GCNFT |
|---|---|---|---|---|---|---|---|---|---|
| MLP | 75.63 | 80.97 | 68.53 | 77.58 | 78.81 | 82.03 | 84.86 | 90.70 | **94.01** |
| KNN | 85.25 | 93.38 | 80.93 | 92.43 | 93.24 | 93.40 | 93.70 | 95.93 | **96.96** |
| SVM | 70.89 | 73.91 | 68.56 | 83.82 | 82.73 | 80.52 | 84.86 | 84.53 | **85.89** |
| LASSO | 70.91 | 84.75 | 68.58 | 84.90 | 85.33 | 84.33 | 85.88 | 87.19 | **89.97** |
| Ridge | 70.87 | 78.35 | 68.58 | 78.58 | 82.68 | 84.56 | 82.69 | 83.44 | **85.94** |

**A Study of GCN-Awareness.** This experiment aims to answer: *Does the proposed method have GCN awareness?* To study the GCN-awareness of GCNFT, we leverage t-SNE to visualize the original feature space and the feature spaces transformed by different methods on the graph classification task of PRO-TEINS_full. We use different colors to denote different subgraphs of the dataset and blue ellipses are utilized to highlight the limitations of the baselines, i.e., there is significant overlap among the nodes of different subgraphs. As shown in Figure 6, even for the two best-performing baselines, there is still a clear overlap in the node distribution across different subgraphs. This is because both GRFG and MOAT can only perform feature transformations on tabular datasets. Without prior exposure to the graph structure, it is difficult for them to distinguish between the nodes of different subgraphs. However, the node distribution of GCNFT is discriminative, compared with the baselines. We can observe that the nodes within the same subgraph tend to cluster together. This implies that the graph convolutional structure score effectively simulates the node aggregation process of GCN. Therefore, we can confirm that GCNFT has GCN awareness.

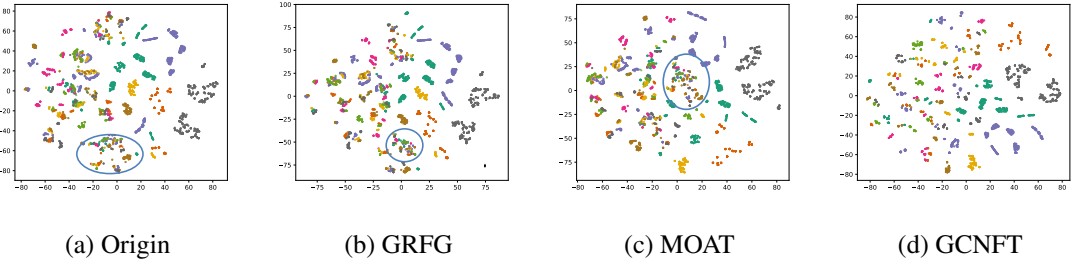

(a) Origin                (b) GRFG                (c) MOAT                (d) GCNFT

Figure 6: The t-SNE distribution visualization of the original feature space and the transformed feature spaces obtained from different methods on PROTEINS_full. Different colors denotes different subgraphs. We highlight the limitations of the baselines using blue ellipses.

**Visualization Analysis of The Learned Embedding Space**. This experiment aims to answer: *Is the latent embedding space well-constructed and helpful for gradient-steered search?* Taking graph classification tasks as an example, we study the latent embedding space. We use t-SNE to visualize the learned embedding space. Each point denotes a transformed attributed graph, where the point is darker, the performance is better. We can observe that the top-performance points tend to be close in the latent embedding space. The possible reason is

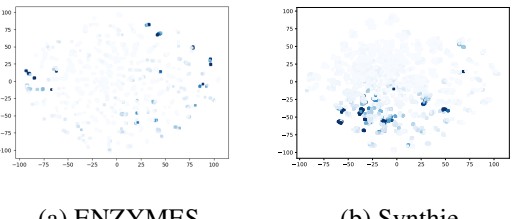

(a) ENZYMES          (b) Synthie

Figure 7: The t-SNE distribution visualization of the learned embedding space. Darker points indicate feature transformations with better performance.

that transformed attribute graphs with higher downstream task performance are likely clustered in a certain region, which enables the gradient-search method to effectively and easily locate the best result within these areas when we use these strong data points as our initial search seeds.

## 5 RELATED WORK

**Feature Transformation.** Feature transformation aims to reconstruct the feature space by transforming features by mathematical operations (Gong et al., 2024; Wang et al., 2022b; Ying et al., 2024; Blei et al., 2003a; Horn et al., 2020a). The existing works are mainly three folds: 1) Expansion-reduction methods (Kanter & Veeramachaneni, 2015; Horn et al., 2020b; Khurana et al., 2016). This kind of methods firstly generate new features by transforming original features to expand the feature space. Then, they perform feature selection to eliminate redundant features and retain useful features. However, it is hard for these methods to generate complex transformations. Also, since the search space increases exponentially, the efficiency is limited. 2) Evolution-evaluation methods (Khurana et al., 2018; Wang et al., 2022b; Xiao et al., 2023a; Tran et al., 2016). To effectively and efficiently search the feature transformation space, methods like genetic programming and reinforcement learning are introduced to this field. They iteratively generate new features and optimized according to the feedback of downstream ML models. Compared with expansion-reduction methods, the optimized strategies facilitate more robust and stable search. However, the nature of decision-making process in the discrete space makes these methods still time-consuming and tend to fail into local optima. 3) NAS-based methods (Chen et al., 2019; Zhu et al., 2022). NAS is proposed to search the model architecture automatically (Li et al., 2021; Elsken et al., 2019). The success of NAs in many areas (Wang et al., 2021; Xiao et al., 2023b; Wever et al., 2021) inspires the application in feature transformation. However, NAS-based methods fails to generate high-order feature transformations and the performance is not stable. The prior literature mainly focus on tabular feature transformation. As graphs become a crucial data structure to represent complex relationships, we propose a novel GCN-aware feature transformation method to solve the significant graph feature transformation task.

**Graph Neural Network.** Graph data is prevalent in the real world. Neural networks were first applied to modeling directed acyclic graphs in (Sperduti & Starita, 1997), which laid the foundation for the early design and development of Graph Neural Networks (GNNs) (Gori et al., 2005; Scarselli et al., 2008; Gallicchio & Micheli, 2010). Building upon this, the success of Convolutional Neural Networks (CNNs) in computer vision inspired researchers to explore the application of convolutional operations to graph data. Graph Convolutional Neural Networks (GCNs) (Bruna et al., 2013; Kipf & Welling, 2016) redefined convolution operations on graphs and have since attracted considerable attention from both academia and industry. Subsequent research has explored various methods to improve the aggregation of information in GCNs (Veličković et al., 2017; Xu et al., 2018), while others have sought to enhance the structural complexity of the models (Li et al., 2019; Pei et al., 2022). Moreover, some studies have focused on addressing the over-smoothing problem that occurs in deeper GCN architectures, which limits their potential (Du et al., 2018; Hu et al., 2019; Chen et al., 2020). Despite these advancements, traditional graph convolution methods often require significant computational resources, rendering them difficult to scale to large graphs. To mitigate this, several approaches have been proposed to simplify GCNs from different perspectives (Dai et al., 2018; Gu et al., 2020; Liu et al., 2020; Wu et al., 2019). Specifically, (Dai et al., 2018) andGu et al. (2020) extend fixed-point theory in GNNs to improve representation learning. Motivated by these simplified GCN architectures, we propose a novel graph structure-aware feature transformation method that is computationally efficient.

## 6 CONCLUSION

We studied the problem of GCNFT: GCN-aware feature transformation and developed a knowledge guided generative learning perspective to integrate GCN awareness into generative transformation. To quantify GCN awareness, we developed a generic task-agnostic approximate: graph convolutional structure score for loss regularization. To bridges GCN awareness and feature transformation, we developed a graph convolutional structure score guided encoder-evaluator-decoder approach for generative feature transformation. Extensive experiments on node, link, and graph prediction tasks validated the effectiveness of our approach, demonstrating superior performance compared to traditional feature transformation methods. Future work will explore more generalized and effective methods to approximate and integrate other graph structural awareness that goes beyond graph local convolutions.

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
