# OpenReview forum: "GCNFT: Graph Convolutional Networks Aware Generative Feature Transformation"
_ICLR.cc/2025/Conference — Submitted to ICLR 2025_

### Official Review · Reviewer_CPod · 2024-10-30

**Soundness:** 2
**Presentation:** 1
**Contribution:** 2
**Rating:** 3
**Confidence:** 3

**Summary:**

The authors focus on addressing the problem of GCN-aware Feature Transformation (GCNFT) on attributed graphs. They proposed a method comprising two main components. In the first part, the authors analyze existing GCNs and introduce an approach to quantify GCN awareness by generating a score that measures the L2 distance between ideal node representations and the generated representations. In the second part, they present an autoencoder-based method that uses this score as guidance to generate new graph features. The authors claim to have conducted extensive experiments to validate the effectiveness of the proposed method.

**Strengths:**

1. In graph representation learning area, the proposed method is interesting and novel.
2. The code is provided, which is helpful to reproduce the results and verify the ideas in this paper.

**Weaknesses:**

1. In my opinion, the most serious issue is that the authors appear to have limited knowledge of the field of graph representation learning (GRL). The GCNFT problem proposed in Equation 1 seems essentially equivalent to graph representation learning. So, this paper seems simply combine methods in feature transformation with graph, without without sufficient investigation of GRL methods. The proposed method should be both theoretically and experimentally compared with recent advanced GRL methods rather than traditional preprocessing methods that do not even focus on graph data.
2. I strongly recommend that this paper should be reorganized. The current version contains numerous unclear statements that are difficult to understand. The motivation is not well explained, and many parts of the proposed method lack clear definition or explanation. Additionally, appropriate citations should be added in sections other than the related work.
3. In my opinion, the experiments are insufficient and fail to validate the proposed method. Specifically:
> - The paper does not provide enough details about the experimental settings, such as the split ratio of the train, valid and test sets, which is crucial since most GRL methods are semi-supervised.
> - The comparison algorithms should include state-of-the-art GRL methods, yet the paper does not even include GCN (which is a component of the proposed method).
> - The datasets are limited, as the experiments only involve three small-scale graph datasets.
> - There is no complexity analysis, which is important because the authors claim in Section 5 that the proposed method is computationally efficient compared to GNNs.

**Questions:**

Please see above. Additionally, my primary research focus is on graph representation learning, so I am not very familiar with the field of feature transformation. If you can address my concerns, I would be happy to reconsider and adjust my score.

**Details Of Ethics Concerns:**

NA.

---

### Official Review · Reviewer_RLHm · 2024-10-31

**Soundness:** 2
**Presentation:** 3
**Contribution:** 2
**Rating:** 3
**Confidence:** 4

**Summary:**

This paper proposes a method called GCNFT (GCN-aware Feature Transformation).  This framework aims to tackle the challenge of transforming raw node attributes in attributed graphs into more informative features while preserving graph structural information. The proposed method identifies and addresses two key challenges: quantifying GCN awareness and bridging it with feature transformation. The authors propose a task-agnostic structure score derived from interpreting GCN as a gap minimization process in Laplacian smoothing and develop a generative learning framework that uses this score to guide feature transformation. The framework models feature transformation as a sequence generation task with a two-stage optimization process combining structural and performance objectives.

**Strengths:**

S1. Motivations: The concept of "how to generate explicit, interpretable feature transformations while preserving graph structural information" is interesting.
S2. Framework: The multi-agent reinforcement learning framework for feature exploration (head, operation, and tail agents) offers a systematic approach to discovering complex feature transformations while maintaining graph awareness through the structure-guided reward.
S3. Experiments: The empirical results do show consistent improvements over baselines.

**Weaknesses:**

W1. The major concern is technical novelty: the sequence generation approach relies on basic LSTM architectures. The claimed "generative AI" aspect is essentially just sequence-to-sequence learning with LSTM. The reinforcement learning exploration strategy lacks theoretical guarantees.
W2. The empirical validation is inadequate for the paper's claims. The datasets are extremely small (the largest has only 1113 graphs with an average of 39 nodes), and there's no evaluation on real-world large graphs. The reported 3-20% improvements lack proper statistical significance analysis.
W3. I believe some critical implementation details are missing. The feature vocabulary construction, handling of invalid expressions, and numerical stability measures are not specified. The complexity analysis of structure score computation is absent.
W4. There is also a line of works [1,2] using LLM augmenting graph attributes which is totally neglected in this paper.

[1] Harnessing Explanations: LLM-to-LM Interpreter for Enhanced Text-Attributed Graph Representation Learning.
[2] G-retriever: Retrieval-augmented generation for textual graph understanding and question-answering.

**Questions:**

Q1. The structure score computation appears to be O(|V|^2) for dense graphs. How does this scale to large graphs? Have you considered approximation techniques for better efficiency?
Q2. The feature generation process lacks crucial details. How do you ensure the mathematical validity of generated features? How do you handle operator precedence and prevent feature explosion?
Q3. The experiments lack comparison with recent graph learning methods. How does GCNFT compare with graph foundation models or more recent GNNs that also aim to capture structural information?
Q4. The multi-agent RL framework's training details are unclear. What is the exploration strategy? How do you coordinate between agents? Can you provide examples of discovered feature patterns?
Q5. The paper claims improved interpretability but provides no analysis of the generated features' semantic meaning or utility. Can you demonstrate how the transformed features are more interpretable than GCN embeddings?

---

### Official Review · Reviewer_USH6 · 2024-11-04

**Soundness:** 2
**Presentation:** 3
**Contribution:** 2
**Rating:** 3
**Confidence:** 4

**Summary:**

This paper proposes the problem of GCN-aware Feature Transformation and clarifies two challenges behind the problem: (1) quantifying GCN awareness and (2) bridging GCN awareness and feature transformation. The authors propose a method to resolve the GCN-aware feature transformation task for attributed graphs. Firstly, the proposed method quantifies GCN awareness by approximating the GCN into a graph convolutional structure score as a regularization term. Secondly, they develop a score-guided generative feature transformation approach that consists of embedding, optimization, and generation. Extensive experiments are conducted on node classification, link prediction and graph classification on three datasets. Results show that the proposed GCNFT outperforms other feature transformation baselines in general.

**Strengths:**

- The paper is well-written and easy to follow.
- The motivation for incorporating structural awareness into feature transformation is reasonable and novel to me.
- The experimental results are significant and promising.

**Weaknesses:**

- The motivation for sequential modeling of transformed features needs stronger justification. While the success of LLMs is noted (Lines 65-66), the connection between LLM's generative capabilities and the proposed sequential feature tokens requires a clearer theoretical explanation, as I cannot see a straightforward rationale for conditional feature generation.
- The reinforcement learning component would benefit from more detailed explanation, including the specific architecture, reward function design, and training procedure.
- The choice of LSTM for sequential modeling warrants comparison with modern transformer-based architectures. Additional experiments comparing the performance of LSTM versus transformer variants would strengthen the technical contribution.
- The use of GCNs for heterogeneous graphs raises concerns about the model's ability to capture complex relationships between different node types. Consider discussing potential limitations or justifying this architectural choice.
-  Complexity and scalability may be the concern of the proposed method. The method involves the curation of a training dataset from the RL system, complicated two-stage optimization with evaluators, and a generative decoding process, which may not be scalable to large-scale graph datasets with high-dimensional raw features. Empirical results should be included to justify the algorithm's complexity.
- GNN-based baselines and standard deviations are missing in all result tables.

Minor Comments:
1. Consider adding ablation studies to validate key components
2. Include runtime comparisons with baseline methods
3. Clarify the hyperparameter selection process

**Questions:**

- Authors should provide more discussions and clarification of the reinforcement learning system. How are the feature selection agents designed and implemented? What is the specific formulation of the reward function?
- GCN is usually constrained by over-smoothing issues when stacking multiple message-passing layers. Equation (2) is essentially the convergence of node representation caused by over-smoothing, where the node representations are getting more and more indistinguishable. It is unclear how the Equation 2 is related to the over-smoothing issue. I also suspect the effectiveness of the ideal node representation derived from Equation (2).
- Some baselines are missing. The authors simply compare with traditional feature transformations. I am curious about the performance difference from state-of-the-art GNN methods. Is there any reason for not comparing with GNN-based baselines?
- For Figure 7, how do structure scores correlate with downstream task performance? How are step lengths ($\alpha$ and $\beta$) tuned to avoid local minima, particularly in clustered regions?

---

### Official Review · Reviewer_3Lcd · 2024-11-04

**Soundness:** 2
**Presentation:** 2
**Contribution:** 2
**Rating:** 3
**Confidence:** 4

**Summary:**

This manuscript introduces a novel framework for feature transformation in attributed graphs by incorporating GCN (Graph Convolutional Network) awareness. The primary goal is to transform raw node attributes into enhanced features that maintain both node and structure information. GCNFT imposes two under-addressed challenges: 1) quantifying GCN awareness and 2) bridging GCN awareness and feature transformation. GCNFT improves the performance in downstream tasks such as node classification, link prediction, and graph classification.

**Strengths:**

The paper addresses an innovative problem of integrating GCN awareness into feature transformation, bridging the gap between traditional feature engineering and GCN-based approaches.
The visualization shows the advantage of the proposed method compared with baselines.

**Weaknesses:**

1. The symbols are quite inconsistent in the manuscript. For example both A and bold(A) represent the adjacency matrix. Some of the matrix and vectors are in boldface, while some others don't.
2. No theoretical analysis of how the proposed graph convolutional structure score can ensure the model learns a better representation compared to conventional GCN methods.
3. The experiments are quite simple. It is doubtful if the proposed GCNFT can be applied to more complicated graphs such as OGNB large datasets.  https://ogb.stanford.edu/docs/nodeprop/
4. Computing graph structure score seems quite complicated for large-scale graphs. A study on the complicity needs to be analyzed.

**Questions:**

Please refer to Weaknesses.

---

### Meta-Review · Area_Chair_Tg7V · 2024-12-07

**Metareview:**

The submission introduces a novel framework to enhance feature representation for attributed graphs by incorporating structural awareness through a GCN-inspired structure score. The framework addresses two key challenges: quantifying GCN awareness and connecting it to feature transformation. The primary strength lies in the innovative problem formulation. However, several concerns limit the impact of the work, including the use of limited datasets for evaluation, the absence of critical baselines, the lack of advanced sequence modeling techniques such as transformers, insufficient theoretical justification, and unclear notations and organization. Given the unanimous rejection by all reviewers and the highlighted weaknesses, I recommend rejecting the submission.

**Additional Comments On Reviewer Discussion:**

N/A

---

### Decision · Program_Chairs · 2025-01-22

Reject